# Fast Classification Rates for High-dimensional Gaussian Generative Models

**Tianyang Li**                **Adarsh Prasad**                **Pradeep Ravikumar**

Department of Computer Science, UT Austin

{lty,adarsh,pradeepr}@cs.utexas.edu

## Abstract

We consider the problem of binary classification when the covariates conditioned on the each of the response values follow multivariate Gaussian distributions. We focus on the setting where the covariance matrices for the two conditional distributions are the same. The corresponding generative model classifier, derived via the Bayes rule, also called Linear Discriminant Analysis, has been shown to behave poorly in high-dimensional settings. We present a novel analysis of the classification error of any linear discriminant approach given conditional Gaussian models. This allows us to compare the generative model classifier, other recently proposed discriminative approaches that directly learn the discriminant function, and then finally logistic regression which is another classical discriminative model classifier. As we show, under a natural sparsity assumption, and letting $s$ denote the sparsity of the Bayes classifier, $p$ the number of covariates, and $n$ the number of samples, the simple ($\ell_1$-regularized) logistic regression classifier achieves the fast misclassification error rates of $O\left(\frac{s \log p}{n}\right)$, which is much better than the other approaches, which are either inconsistent under high-dimensional settings, or achieve a slower rate of $O\left(\sqrt{\frac{s \log p}{n}}\right)$.

## 1  Introduction

We consider the problem of classification of a binary response given $p$ covariates. A popular class of approaches are statistical decision-theoretic: given a classification evaluation metric, they then optimize a surrogate evaluation metric that is computationally tractable, and yet have strong guarantees on sample complexity, namely, number of observations required for some bound on the expected classification evaluation metric. These guarantees and methods have been developed largely for the zero-one evaluation metric, and extending these to general evaluation metrics is an area of active research. Another class of classification methods are relatively evaluation metric agnostic, which is an important desideratum in modern settings, where the evaluation metric for an application is typically less clear: these are based on learning statistical models over the response and covariates, and can be categorized into two classes. The first are so-called generative models, where we specify conditional distributions of the covariates conditioned on the response, and then use the Bayes rule to derive the conditional distribution of the response given the covariates. The second are the so-called discriminative models, where we directly specify the conditional distribution of the response given the covariates.

In the classical fixed $p$ setting, we have now have a good understanding of the performance of the classification approaches above. For generative and discriminative modeling based approaches, consider the specific case of Naive Bayes generative models and logistic regression discriminative models (which form a so-called generative-discriminative pair[1]), Ng and Jordan [27] provided qual-

itative consistency analyses, and showed that under small sample settings, the generative model classifiers converge at a faster rate to their population error rate compared to the discriminative model classifiers, though the population error rate of the discriminative model classifiers could be potentially lower than that of the generative model classifiers due to weaker model assumptions. But if the generative model assumption holds, then generative model classifiers seem preferable to discriminative model classifiers.

In this paper, we investigate whether this conventional wisdom holds even under high-dimensional settings. We focus on the simple generative model where the response is binary, and the covariates conditioned on each of the response values, follows a conditional multivariate Gaussian distribution. We also assume that the two covariance matrices of the two conditional Gaussian distributions are the same. The corresponding generative model classifier, derived via the Bayes rule, is known in the statistics literature as the Linear Discriminant Analysis (LDA) classifier [21]. Under classical settings where $p \ll n$, the misclassification error rate of this classifier has been shown to converge to that of the Bayes classifier. However, in a high-dimensional setting, where the number of covariates $p$ could scale with the number of samples $n$, this performance of the LDA classifier breaks down. In particular, Bickel and Levina [3] show that when $p/n \to \infty$, then the LDA classifier could converge to an error rate of 0.5, that of random chance. What should one then do, when we are even allowed this generative model assumption, and when $p > n$?

Bickel and Levina [3] suggest the use of a Naive Bayes or conditional independence assumption, which in the conditional Gaussian context, assumes the covariance matrices to be diagonal. As they showed, the corresponding Naive Bayes LDA classifier does have misclassification error rate that is better than chance, but it is asymptotically biased: it converges to an error rate that is strictly larger than that of the Bayes classifier when the Naive Bayes conditional independence assumption does not hold. Bickel and Levina [3] also considered a weakening of the Naive Bayes rule, by assuming that the covariance matrix is weakly sparse, and an ellipsoidal constraint on the means, showed that an estimator that leverages these structural constraints converges to the Bayes risk at a rate of $O(\log(n)/n^\gamma)$, where $0 < \gamma < 1$ depends on the mean and covariance structural assumptions. A caveat is that these covariance sparsity assumptions might not hold in practice. Similar caveats apply to the related works on feature annealed independence rules [14], nearest shrunken centroids [29, 30], as well as those . Moreover, even when the assumptions hold, they do not yield the "fast" rates of $O(1/n)$.

An alternative approach is to directly impose sparsity on the linear discriminant [28, 7], which is weaker than the covariance sparsity assumptions (though [28] impose these in addition). [28, 7] then proposed new estimators that leveraged these assumptions, but while they were able to show convergence to the Bayes risk, they were only able to show a slower rate of $O\left(\sqrt{\frac{s \log p}{n}}\right)$.

It is instructive at this juncture to look at recent results on classification error rates from the machine learning community. A key notion of importance here is whether the two classes are separable: which can be understood as requiring that the classification error of the Bayes classifier is 0. Classical learning theory gives a rate of $O(1/\sqrt{n})$ for any classifier when two classes are non-separable, and it shown that this is also minimax [12], with the note that this is relatively distribution agnostic, since it assumes very little on the underlying distributions. When the two classes are non-separable, only rates slower than $\Omega(1/n)$ are known. Another key notion is a "low-noise condition" [25], under which certain classifiers can be shown to attain a rate faster than $o(1/\sqrt{n})$, albeit not at the $O(1/n)$ rate unless the two classes are separable. Specifically, let $\alpha$ denote a constant such that

$$\mathbb{P}\left(|\mathbb{P}(Y=1|X) - 1/2| \le t\right) \le O\left(t^\alpha\right), \tag{1}$$

holds when $t \to 0$. This is said to be a low-noise assumption, since as $\alpha \to +\infty$, the two classes start becoming separable, that is, the Bayes risk approaches zero. Under this low-noise assumption, known rates for excess 0-1 risk is $O\left(\left(\frac{1}{n}\right)^{\frac{1+\alpha}{2+\alpha}}\right)$ [23]. Note that this is always slower than $O(\frac{1}{n})$ when $\alpha < +\infty$.

There has been a surge of recent results on high-dimensional statistical statistical analyses of $M$-estimators [26, 9, 1]. These however are largely focused on parameter error bounds, empirical and population log-likelihood, and sparsistency. In this paper however, we are interested in analyzing the *zero-one classification error* under high-dimensional sampling regimes. One could stitch these recent results to obtain some error bounds: use bounds on the excess log-likelihood, and use trans-

forms from [2], to convert excess log-likelihood bounds to get bounds on 0-1 classification error, however, the resulting bounds are very loose, and in particular, do not yield the fast rates that we seek.

In this paper, we leverage the closed form expression for the zero-one classification error for our generative model, and directly analyse it to give faster rates for any linear discriminant method. Our analyses show that, assuming a sparse linear discriminant in addition, the simple $\ell_1$-regularized logistic regression classifier achieves near optimal fast rates of $O\left(\frac{s \log p}{n}\right)$, even without requiring that the two classes be separable.

## 2 Problem Setup

We consider the problem of high dimensional binary classification under the following generative model. Let $Y \in \{0, 1\}$ denote a binary response variable, and let $X = (X_1, \ldots, X_p) \in \mathbb{R}^p$ denote a set of $p$ covariates. For technical simplicity, we assume $\Pr[Y = 1] = \Pr[Y = 0] = \frac{1}{2}$, however our analysis easily extends to the more general case when $\Pr[Y = 1], \Pr[Y = 0] \in [\delta_0, 1 - \delta_0]$, for some constant $0 < \delta_0 < \frac{1}{2}$. We assume that $X|Y \sim \mathcal{N}(\mu_Y, \Sigma_Y)$, i.e. conditioned on a response, the covariate follows a multivariate Gaussian distribution. We assume we are given $n$ training samples $\{(X^{(1)}, Y^{(1)}), (X^{(2)}, Y^{(2)}), \ldots, (X^{(n)}, Y^{(n)})\}$ drawn i.i.d. from the conditional Gaussian model above.

For any classifier, $C : \mathbb{R}^p \to \{1, 0\}$, the *0-1 risk* or simply the *classification error* is given by $R_{0-1}(C) = \mathbb{E}_{X,Y}[\ell_{0-1}(C(X), Y)]$, where $\ell_{0-1}(C(x), y) = \mathbf{1}(C(x) \neq y)$ is the 0-1 loss. It can also be simply written as $R(C) = \Pr[C(X) \neq Y]$. The classifier attaining the lowest classification error is known as the Bayes classifier, which we will denote by $C^*$. Under the generative model assumption above, the Bayes classifier can be derived simply as $C^*(X) = \mathbf{1}(\log \frac{\Pr[Y=1|X]}{\Pr[Y=0|X]} > 0)$, so that given sample $X$, it would be classified as 1 if $\frac{\Pr[Y=1|X]}{\Pr[Y=0|X]} > 1$, and as 0 otherwise. We denote the error of the Bayes classifier $R^* = R(C^*)$.

When $\Sigma_1 = \Sigma_0 = \Sigma$,

$$\log \frac{\Pr[Y=1|X]}{\Pr[Y=0|X]} = (\mu_1 - \mu_0)^T \Sigma^{-1} X + \frac{1}{2}(-\mu_1^T \Sigma^{-1} \mu_1 + \mu_0^T \Sigma^{-1} \mu_0) \tag{2}$$

and we denote this quantity as $w^{*T} X + b^*$ where

$$w^* = \Sigma^{-1}(\mu_1 - \mu_0), \ b^* = \frac{-\mu_1^T \Sigma^{-1} \mu_1 + \mu_0^T \Sigma^{-1} \mu_0}{2},$$

so that the Bayes classifier can be written as: $C^*(x) = \mathbf{1}(w^{*T} x + b^* > 0)$.

For any trained classifier $\hat{C}$ we are interested in bounding the excess risk defined as $R(\hat{C}) - R^*$. The *generative* approach to training a classifier is to estimate estimate $\Sigma^{-1}$ and $\delta$ from data, and then plug the estimates into Equation 2 to construct the classifier. This classifier is known as the *linear discriminant analysis* (LDA) classifier, whose theoretical properties have been well-studied in classical fixed $p$ setting. The *discriminative* approach to training is to estimate $\frac{\Pr[Y=1|X]}{\Pr[Y=0|X]}$ directly from samples.

### 2.1 Assumptions.

We assume that mean is bounded i.e. $\mu_1, \mu_0 \in \{\mu \in \mathbb{R}^p : \|\mu\|_2 \leq B_\mu\}$, where $B_\mu$ is a constant which doesn't scale with $p$. We assume that the covariance matrix $\Sigma$ is non-degenerate i.e. all eigenvalues of $\Sigma$ are in $[B_{\lambda_{\min}}, B_{\lambda_{\max}}]$. Additionally we assume $(\mu_1 - \mu_0)^T \Sigma^{-1}(\mu_1 - \mu_0) \geq B_s$, which gives a lower bound on the Bayes classifier's classification error $R^* \geq 1 - \Phi(\frac{1}{2}B_s) > 0$. Note that this assumption is different from the definition of separable classes in [11] and the low noise condition in [25], and the two classes are still not separable because $R^* > 0$.

#### 2.1.1 Sparsity Assumption.

Motivated by [7], we assume that $\Sigma^{-1}(\mu_1 - \mu_0)$ is sparse, and there at most $s$ non-zero entries. Cai and Liu [7] extensively discuss and show that such a sparsity assumption, is much weaker than assuming either $\Sigma^{-1}$ and $(\mu_1 - \mu_0)$ to be individually sparse. We refer the reader to [7] for an elaborate discussion.

## 2.2 Generative Classifiers

Generative techniques work by estimating $\Sigma^{-1}$ and $(\mu_1 - \mu_0)$ from data and plugging them into Equation 2. In high-dimensions, simple estimation techniques do not perform well when $p \gg n$, the sample estimate for the covariance matrix $\hat{\Sigma}$ is singular; using the generalized inverse of the sample covariance matrix makes the estimator highly biased and unstable. Numerous alternative approaches have been proposed by imposing structural conditions on $\Sigma$ or $\Sigma^{-1}$ and $\delta$ to ensure that they can be estimated consistently. Some early work based on nearest shrunken centroids [29, 30], feature annealed independence rules [14], and Naive Bayes [4] imposed independence assumptions on $\Sigma$, which are often violated in real-world applications. [4, 13] impose more complex structural assumptions on the covariance matrix and suggest more complicated thresholding techniques. Most commonly, $\Sigma^{-1}$ and $\delta$ are assumed to be sparse and then some thresholding techniques are used to estimate them consistently [17, 28].

## 2.3 Discriminative Classifiers.

Recently, more *direct* techniques have been proposed to solve the sparse LDA problem. Let $\hat{\Sigma}$ and $\hat{\mu_d}$ be consistent estimators of $\Sigma$ and $\mu = \frac{\mu_1 + \mu_0}{2}$. Fan et al. [15] proposed the Regularized Optimal Affine Discriminant (ROAD) approach which minimizes $w^T \Sigma w$ with $w^T \mu$ restricted to be a constant value and an $\ell_1$-penalty of $w$.

$$w_{\text{ROAD}} = \underset{\substack{w^T \hat{\mu}=1 \\ ||w||_1 \leq c}}{\text{argmin}} \, w^T \hat{\Sigma} w \tag{3}$$

Kolar and Liu [22] provided theoretical insights into the ROAD estimator by analysing its consistency for variable selection. Cai and Liu [7] proposed another variant called linear programming discriminant (LPD) which tries to make $w$ close to the Bayes rules linear term $\Sigma^{-1}(\mu_1 - \mu_0)$ in the $\ell_\infty$ norm. This can be cast as a linear programming problem related to the Dantzig selector.[8].

$$w_{\text{LPD}} = \underset{w}{\text{argmin}} \, ||w||_1 \tag{4}$$

$$\text{s.t.} ||\hat{\Sigma} w - \hat{\mu}||_\infty \leq \lambda_n$$

Mai et al. [24]proposed another version of the sparse linear discriminant analysis based on an equivalent least square formulation of the LDA, where they solve an $\ell_1$-regularized least squares problem to produce a consistent classifier.

All the techniques above either do not have finite sample convergence rates, or the *0-1* risk converged at a slow rate of $O\left(\sqrt{\frac{s \log p}{n}}\right)$.

In this paper, we first provide an analysis of classification error rates for any classifier with a linear discriminant function, and then follow this analysis by investigating the performance of generative and discriminative classifiers for conditional Gaussian model.

## 3 Classifiers with Sparse Linear Discriminants

We first analyze any classifier with a linear discriminant function, of the form: $C(x) = \mathbf{1}(w^T x + b > 0)$. We first note that the 0-1 classification error of any such classifier is available in closed-form as

$$R(w, b) = 1 - \frac{1}{2}\Phi\left(\frac{w^T \mu_1 + b}{\sqrt{w^T \Sigma w}}\right) - \frac{1}{2}\Phi\left(-\frac{w^T \mu_0 + b}{\sqrt{w^T \Sigma w}}\right), \tag{5}$$

which can be shown by noting that $w^T X + b$ is a univariate normal random variable when conditioned on the label $Y$.

Next, we relate the 0-1 classifiction error above to that of the Bayes classifier. Recall the earlier notation of the Bayes classifier as $C^*(x) = \mathbf{1}(x^T w^* + b^* > 0)$. The following theorem is a key result of the paper that shows that for any linear discriminant classifier whose linear discriminant parameters are close to that of the Bayes classifier, the excess 0-1 risk is bounded only by second order terms of the difference. Note that this theorem will enable fast classification rates if we obtain fast rates for the parameter error.

**Theorem 1.** *Let $w = w^* + \Delta$, $b = b^* + \Omega$, and $\Delta \to 0$, $\Omega \to 0$, then we have*

$$R(w = w^* + \Delta, b = b^* + \Omega) - R(w^*, b^*) = O(\|\Delta\|_2^2 + \Omega^2). \tag{6}$$

*Proof.* Denote the quantity $S^* = \sqrt{(\mu_1 - \mu_0)^T \Sigma^{-1}(\mu_1 - \mu_0)}$, then we have $\frac{\mu_1^T w^* + b^*}{\sqrt{w^{*T}\Sigma w^*}} = \frac{-\mu_1^T w^* - b^*}{\sqrt{w^{*T}\Sigma w^*}} = \frac{1}{2}S^*$. Using (5) and the Taylor series expansion of $\Phi(\cdot)$ around $\frac{1}{2}S^*$, we have

$$|R(w,b) - R(w^*, b^*)| = \frac{1}{2}|(\Phi(\frac{\mu_1^T w + b}{\sqrt{w^T \Sigma w}}) - \Phi(\frac{1}{2}S^*)) + (\Phi(\frac{-\mu_0^T w - b}{\sqrt{w^T \Sigma w}}) - \Phi(\frac{1}{2}S^*))| \quad (7)$$

$$\leq K_1|\frac{(\mu_1 - \mu_0)^T w}{\sqrt{w^T \Sigma w}} - S^*| + K_2(\frac{\mu_1^T w + b}{\sqrt{w^T \Sigma w}} - \frac{1}{2}S^*)^2 + K_3(\frac{-\mu_0^T w - b}{\sqrt{w^T \Sigma w}} - \frac{1}{2}S^*)^2$$

where $K_1, K_2, K_3 > 0$ are constants because the first and second order derivatives of $\Phi(\cdot)$ are bounded.

First note that $|\sqrt{w^T \Sigma w} - \sqrt{w^{*T}\Sigma w^*}| = O(\|\Delta\|_2)$, because $\|w^*\|_2$ is bounded.

Denote $w'' = \Sigma^{\frac{1}{2}}w$, $\Delta'' = \Sigma^{\frac{1}{2}}\Delta$, $w''^* = \Sigma^{\frac{1}{2}}w^*$ $a'' = \Sigma^{-\frac{1}{2}}(\mu_1 - \mu_0)$, we have (by the binomial Taylor series expansion)

$$\frac{(\mu_1 - \mu_0)^T w}{\sqrt{w^T \Sigma w}} - S^* = \frac{a''^T w''}{\sqrt{w''^T w''}} - \sqrt{a''^T a''} \quad (8)$$

$$= \frac{1 + \frac{a''^T \Delta''}{a''^T a''} - \sqrt{1 + 2\frac{a''^T \Delta''}{a''^T a''} + \frac{\Delta''^T \Delta''}{a''^T a''}}}{\frac{\sqrt{w''^T w''}}{a''^T a''}} = O(\frac{\|\Delta''\|_2^2}{\sqrt{a''^T a''}})$$

Note that $w'' \to a''$, $\Delta'' \to 0$, $\|\Delta\|_2 = \Theta(\|\Delta''\|_2)$, and $S^*$ is lower bouned, we have $|\frac{(\mu_1 - \mu_0)^T w}{\sqrt{w^T \Sigma w}} - S^*| = O(\|\Delta\|_2^2)$.

Next we bound $|\frac{\mu_1^T w + b}{\sqrt{w^T \Sigma w}} - \frac{1}{2}S^*|$:

$$|\frac{\mu_1^T w + b}{\sqrt{w^T \Sigma w}} - \frac{1}{2}S^*| = |\frac{(\mu_1^T w^* + b^*)(\sqrt{w^{*T}\Sigma w^*} - \sqrt{w^T \Sigma w}) + \sqrt{w^{*T}\Sigma w^*}(\mu_1^T \Delta + \Omega)}{\sqrt{w^T \Sigma w}\sqrt{w^{*T}\Sigma w^*}}| \quad (9)$$

$$= O(\sqrt{\|\Delta\|_2^2 + \Omega^2})$$

where we use the fact that $|\mu_1^T w^* + b^*|$ and $S^*$ are bounded.

Similarly $|\frac{-\mu_0^T w - b}{\sqrt{w^T \Sigma w}} - \frac{1}{2}S^*| = O(\sqrt{\|\Delta\|_2^2 + \Omega^2})$.

Combing the above bounds we get the desired result. $\square$

# 4 Logistic Regression Classifier

In this section, we show that the simple $\ell_1$ regularized logistic regression classifier attains fast classification error rates.

Specifically, we are interested in the $M$-estimator [21] below:

$$(\hat{w}, \hat{b}) = \arg\min_{w,b}\left\{\frac{1}{n}\sum(Y^{(i)}(w^T X^{(i)} + b) + \log(1 + \exp(w^T X^{(i)} + b))) + \lambda(\|w\|_1 + |b|)\right\}, \quad (10)$$

which maximizes the penalized log-likelihood of the logistic regression model, which also corresponds to the conditional probability of the response given the covariates $\mathbb{P}(Y|X)$ for the conditional Gaussian model.

Note that here we penalize the intercept term $b$ as well. Although the intercept term usually is not penalized (e.g. [19]), some packages (e.g. [16]) penalize the intercept term. Our analysis show that penalizing the intercept term does not degrade the performance of the classifier.

In [2, 31] it is shown that minimizing the expected risk of the logistic loss also minimizes the classification error for the corresponding linear classifier. $\ell_1$ regularized logistic regression is a popular classification method in many settings [18, 5]. Several commonly used packages ([19, 16]) have been developed for $\ell_1$ regularized logistic regression. And recent works ([20, 10]) have been on scaling regularized logistic regression to ultra-high dimensions and large number of samples.

## 4.1 Analysis

We first show that $\ell_1$ regularized logistic regression estimator above converges to the Bayes classifier parameters using techniques. Next we use the theorem from the previous section to argue that since estimated parameter $\hat{w}$, $\hat{b}$ is close to the Bayes classifier's parameter $w^*$, $b^*$, the excess risk of the classifier using estimated parameter is tightly bounded as well.

For the first step, we first show a restricted eigenvalue condition for $X' = (X, 1)$ where $X$ are our covariates, that comes from a mixture of two Gaussian distributions $\frac{1}{2}\mathcal{N}(\mu_1, \Sigma) + \frac{1}{2}\mathcal{N}(\mu_0, \Sigma)$. Note that $X'$ is not zero centered, which is different from existing scenarios ([26], [6], etc.) that assume covariates are zero centered. And we denote $w' = (w, b)$, $S' = \{i : w'^*_i \neq 0\}$, and $s' = |S'| \leq s+1$.

**Lemma 1.** *With probability $1 - \delta$, $\forall v' \in A' \subseteq \{v' \in \mathbb{R}^{p+1} : \|v'\|_2 = 1\}$, for some constants $\kappa_1, \kappa_2, \kappa_3 > 0$ we have*

$$\frac{1}{n}\|X'v'\|_2 \geq \kappa_1\sqrt{n} - \kappa_2 w(A') - \kappa_3\sqrt{\log\frac{1}{\delta}} \tag{11}$$

*where $w(A') = \mathbb{E}_{g'\sim\mathcal{N}(0, I_{p+1})}[\sup_{a'\in A'} g'^T a']$ is the Gaussian width of $A'$.*

*In the special case when $A' = \{v' : \|v'_{\bar{S'}}\|_1 \leq 3\|v'_{S'}\|_1, \|v'\|_2 = 1\}$, we have $w(A') = O(\sqrt{s\log p})$.*

*Proof.* First note that $X'$ is sub-Gaussian with bounded parameter and

$$\Sigma' = \mathbb{E}[\frac{1}{n}X'^T X'] = \begin{bmatrix} \Sigma + \frac{1}{2}(\mu_1\mu_1^T + \mu_0\mu_0^T) & \frac{1}{2}(\mu_1 + \mu_0) \\ \frac{1}{2}(\mu_1^T + \mu_0^T) & 1 \end{bmatrix} \tag{12}$$

Note that $A\Sigma' A^T = \begin{bmatrix} \Sigma + \frac{1}{4}(\mu_1 - \mu_0)^T(\mu_1 - \mu_0) & 0 \\ 0 & 1 \end{bmatrix}$ where $A = \begin{bmatrix} I_p & -\frac{1}{2}(\mu_1 + \mu_0) \\ 0 & 1 \end{bmatrix}$, and $A^{-1} = \begin{bmatrix} I_p & \frac{1}{2}(\mu_1 + \mu_0) \\ 0 & 1 \end{bmatrix}$. Notice that $AA^T = \begin{bmatrix} I_p & 0 \\ 0 & 0 \end{bmatrix} + \begin{bmatrix} -\frac{1}{2}(\mu_1 + \mu_0) \\ 1 \end{bmatrix} \begin{bmatrix} -\frac{1}{2}(\mu_1 + \mu_0)^T & 1 \end{bmatrix}$ and $A^{-1}A^{-T} = \begin{bmatrix} I_p & 0 \\ 0 & 0 \end{bmatrix} + \begin{bmatrix} \frac{1}{2}(\mu_1 + \mu_0) \\ 1 \end{bmatrix} \begin{bmatrix} \frac{1}{2}(\mu_1 + \mu_0)^T & 1 \end{bmatrix}$, and we can see that the singular values of $A$ and $A^{-1}$ are lower bounded by $\frac{1}{\sqrt{2 + B_\mu^2}}$ and upper bounded by $\sqrt{2 + B_\mu^2}$. Let $\lambda_1$ be the minimum eigenvalue of $\Sigma'$, and $u'_1$ ($\|u'_1\|_2 = 1$) the corresponding eigenvector. From the expression $A\Sigma A^T A^{-T} u'_1 = \lambda_1 A u'_1$, so we know that the minimum eigenvalue of $\Sigma'$ is lower bounded. Similarly the largest eigenvalue of $\Sigma'$ is upper bounded. Then the desired result follows the proof of Theorem 10 in [1]. Although the proof of Theorem 10 in [1] is for zero-centered random variables, the proof remains valid for non zero-centered random variables.

When $A' = \{v' : \|v'_{\bar{S'}}\|_1 \leq 3\|v'_{S'}\|_1, \|v'\|_2 = 1\}$, [9] gives $w(A') = O(\sqrt{s\log p})$. $\qquad\square$

Having established a restricted eigenvalue result in Lemma 1, next we use the result in [26] for parameter recovery in generalized linear models (GLMs) to show that $\ell_1$ regularized logistic regression can recover the Bayes classifier parameters.

**Lemma 2.** *When the number of samples $n \gg s'\log p$, and choose $\lambda = c_0\sqrt{\frac{\log p}{n}}$ for some constant $c_0$, then we have*

$$\|w^* - \hat{w}\|_2^2 + (b^* - \hat{b})^2 = O(\frac{s'\log p}{n}) \tag{13}$$

*with probability at least $1 - O(\frac{1}{p^{c_1}} + \frac{1}{n^{c_2}})$, where $c_1, c_2 > 0$ are constants.*

*Proof.* Following the proof of Lemma 1, we see that the conditions **(GLM1)** and **(GLM2)** in [26] are satisfied. Following the proof of Proposition 2 and Corollary 5 in [26], we have the desired result. Although the proof of Proposition 2 and Corollary 5 in [26] is for zero-centered random variables, the proof remains valid for non zero-centered random variables. $\qquad\square$

Combining Lemma 2 and Theorem 1 we have the following theorem which gives a fast rate for the excess 0-1 risk of a classifier trained using $\ell_1$ regularized logistic regression.

**Theorem 2.** *With probability at least $1 - O(\frac{1}{p^{c_1}} + \frac{1}{n^{c_2}})$ where $c_1, c_2 > 0$ are constants, when we set $\lambda = c_0\sqrt{\frac{\log p}{n}}$ for some constant $c_0$ the Lasso estimate $\hat{w}, \hat{b}$ in (10) satisfies*

$$R(\hat{w}, \hat{b}) - R(w^*, b^*) = O(\frac{s \log p}{n}) \tag{14}$$

*Proof.* This follows from Lemma 2 and Theorem 1. $\qquad\square$

## 5 Other Linear Discriminant Classifiers

In this section, we provide convergence results for the 0-1 risk for other linear discriminant classifiers discussed in Section 2.3.

**Naive Bayes** We compare the discriminative approach using $\ell_1$ logistic regression to the generative approach using naive Bayes. For illustration purposes we conside the case where $\Sigma = I_p$, $\mu_1 = \frac{M_1}{\sqrt{s}}\begin{bmatrix} 1_s \\ 0_{p-s} \end{bmatrix}$ and $\mu_0 = -\frac{M_0}{\sqrt{s}}\begin{bmatrix} 1_s \\ 0_{p-s} \end{bmatrix}$. where $0 < B_1 \leq M_1, M_0 \leq B_2$ are unknown but bounded constants. In this case $w^* = \frac{M_1+M_0}{\sqrt{s}}\begin{bmatrix} 1_s \\ 0_{p-s} \end{bmatrix}$ and $b^* = \frac{1}{2}(-M_1^2 + M_0^2)$. Using naive Bayes we estimate $\hat{w} = \bar{\mu}_1 - \bar{\mu}_0$, where $\bar{\mu}_1 = \frac{1}{\sum \mathbf{1}(Y^{(i)}=1)})\sum_{Y^{(i)}=1} X^{(i)}$ and $\bar{\mu}_0 = \frac{1}{\sum \mathbf{1}(Y^{(i)}=0)})\sum_{Y^{(i)}=0} X^{(i)}$. Thus with high probability, we have $\|\hat{w} - w^*\|_2^2 = O(\frac{p}{n})$, using Theorem 1 we get a slower rate than the bound given in Theorem 2 for discriminative classification using $\ell_1$ regularized logistic regression.

**LPD [7]** LPD uses a linear programming similar to the Dantzig selector.

**Lemma 3** (Cai and Liu [7], Theorem 4). *Let $\lambda_n = C\sqrt{\frac{s \log p}{n}}$ with $C$ being a sufficiently large constant. Let $n > \log p$, let $\Delta = (\mu_1 - \mu_0)^T \Sigma^{-1}(\mu_1 - \mu_0) > c_1$ for some constant $c_1 > 0$, and let $w_{LPD}$ be obtained as in Equation 4, then with probability greater than $1 - O(p^{-1})$, we have $\frac{R(w_{LPD})}{R^*} - 1 = O(\sqrt{\frac{s \log p}{n}})$.*

**SLDA [28]** SLDA uses thresholded estimate for $\Sigma$ and $\mu_1 - \mu_0$. We state a simpler version.

**Lemma 4** ([28], Theorem 3). *Assume that $\Sigma$ and $\mu_1 - \mu_0$ are sparse, then we have $\frac{R(w_{SLDA})}{R^*} - 1 = O(\max((\frac{s \log p}{n})^{\alpha_1}, (\frac{S \log p}{n})^{\alpha_2}))$ with high probability, where $s = \|\mu_1 - \mu_0\|_0$, $S$ is the number of non-zero entries in $\Sigma$, and $\alpha_1, \alpha_2 \in (0, \frac{1}{2})$ are constants.*

**ROAD [15]** ROAD minimizes $w^T \Sigma w$ with $w^T \mu$ restricted to be a constant value and an $\ell_1$-penalty of $w$.

**Lemma 5** (Fan et al. [15], Theorem 1). *Assume that with high probability, $\|\hat{\Sigma} - \Sigma\|_\infty = O(\sqrt{\frac{\log p}{n}})$ and $\|\hat{\mu} - \mu\|_\infty = O(\sqrt{\frac{\log p}{n}})$, and let $w_{ROAD}$ be obtained as in Equation 3, then with high probability, we have $R(w_{ROAD}) - R^* = O(\sqrt{\frac{s \log p}{n}})$.*

## 6 Experiments

In this section we describe experiments which illustrate the rates for excess 0-1 risk given in Theorem 2. In our experiments we use Glmnet [19] where we set the option to penalize the intercept term along with all other parameters. Glmnet is popular package for $\ell_1$ regularized logistic regression using coordinate descent methods.

For illustration purposes in all simulations we use $\Sigma = I_p$, $\mu_1 = 1_p + \frac{1}{\sqrt{s}}\begin{bmatrix} 1_s \\ 0_{p-s} \end{bmatrix}$, $\mu_0 = 1_p - \frac{1}{\sqrt{s}}\begin{bmatrix} 1_s \\ 0_{p-s} \end{bmatrix}$ To illustrate our bound in Theorem 2, we consider three different scenarios. In Figure 1a

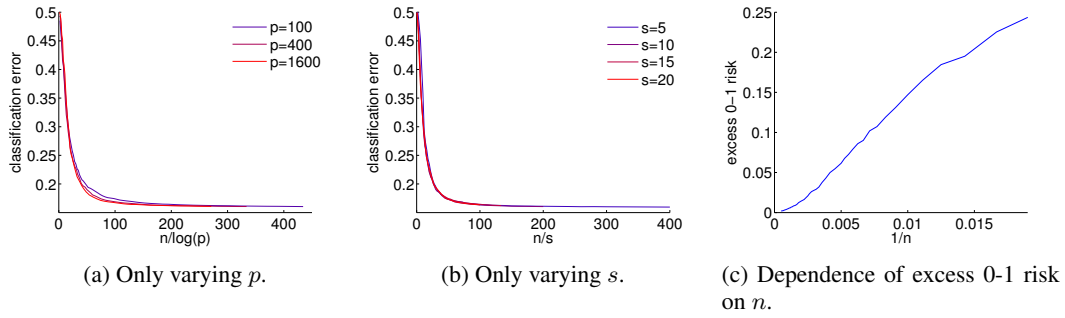

|                    |                    |                    |
|:------------------:|:------------------:|:------------------:|
| (a) Only varying $p$. | (b) Only varying $s$. | (c) Dependence of excess 0-1 risk on $n$. |

Figure 1: Simulations for different Gaussian classification problems showing the dependence of classification error on different quantities. All experiments plotted the average of 20 trials. In all experiments we set the regularization parameter $\lambda = \sqrt{\frac{\log p}{n}}$.

we vary $p$ while keeping $s$, $(\mu_1 - \mu_0)^T \Sigma^{-1} (\mu_1 - \mu_0)$ constant. Figure 1a shows for different $p$ how the classification error changes with increasing $n$. In Figure 1a we show the relationship between the classification error and the quantity $\frac{n}{\log p}$. This figure agrees with our result on excess 0-1 risk's dependence on $p$. In Figure 1b we vary $s$ while keeping $p$, $(\mu_1 - \mu_0)^T \Sigma^{-1} (\mu_1 - \mu_0)$ constant. Figure 1b shows for different $s$ how the classification error changes with increasing $n$. In Figure 1a we show the relationship between the classification error and the quantity $\frac{n}{s}$. This figure agrees with our result on excess 0-1 risk's dependence on $s$. In Figure 1c we show how $R(\hat{w}, \hat{b}) - R(w^*, b^*)$ changes with respect to $\frac{1}{n}$ in one instance Gaussian classification. We can see that the excess 0-1 risk achieves the fast rate and agrees with our bound.

## Acknowledgements

We acknowledge the support of ARO via W911NF-12-1-0390 and NSF via IIS-1149803, IIS-1320894, IIS-1447574, and DMS-1264033, and NIH via R01 GM117594-01 as part of the Joint DMS/NIGMS Initiative to Support Research at the Interface of the Biological and Mathematical Sciences.

## Footnotes

[1]In such a so-called generative-discriminative pair, the discriminative model has the same form as that of the conditional distribution of the response given the covariates specified by the Bayes rule given the generative model

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
