[Reviews · NeurIPS 2015]

Submitted by Assigned_Reviewer_1

-- i don't have many comments, other than to alert the authors to this manuscript: Q. Mai and H. Zou, "A Note On the Connection and Equivalence of Three Sparse Linear Discriminant Analysis Methods," Technometrics, vol. 55, pp. 243-246, 2013. [Online]. Available: http://www.tandfonline.com/doi/abs/10.1080/00401706.2012.746208

which demonstrates that several "different" sparse discriminant algorithms are, in fact, the same. thus, this contribution is perhaps even more useful.
Summary: the authors provide novel misclassification rates under perhaps the simplest high-dimensional classification problem, comparing several different approaches.

Submitted by Assigned_Reviewer_2

+ Well-written paper, claims are clear, + Best convergence rates for the logistic regression classifier under high-dimensional sampling regime, - The usefulness for Machine Learning is left stranded, - The end of the paper is sloppy and finishes quickly while there is room for a conclusion and a discussion.
Summary: This paper presents convergence rates for linear discriminant models and zero-one loss for high-dimensional classification problems. The main result is a convergence rate proportional to the inverse of the training set size, the number of covariates and the sparsity of the Bayes classifier for the logistic regression case (Theorem 2).

Submitted by Assigned_Reviewer_3

I have read the paper ``fast classification rates for high-dimensional conditional Gaussian models".

The paper studies

the problem of binary classification

using a Gaussian model and provides some theoretical results on the convergence of the classification error rates (compared to the Bayes classifier).

The paper presents some nice theoretical results and is interesting to some extent. I am generally positive about the paper but I have the following concerns.

First, it is about the practical relevance.

One of the very important point about classification is that we need a tuning free trained classifier; when a classifier

has tuning parameters that need to be tuned, we usually use data splitting

to decide such tuning parameters. Therefore, it is more relevant to either study

the error rates of a tuning free classifier, or that of a classifier with tuning parameters

selected by data splitting.

This is important for that, while the convergence rate of the "ideal tuning parameters"

may be fast, the convergence rate of the classifier that is actually used could be very slow, simply because of the convergence of the data-driven tuning parameters to the ideal tuning parameters are slow.

In terms of this, the paper needs to (a) compare more carefully with recent tuning-free classifiers, (b) address the effect of selecting the tuning parameters by data-splitting.

Second of all, the paper has put all the focus on the convergence rate,

does not

contain any insightful comparison of the computing speed.

The penalization methods

the authors present are all computationally slow when $p$ is large. It is desirable to

have a detailed comparison about the computational complexity or speed.

Third of all, given this is a well-studied topics, it is desirable to compare

with different methods using at least 1-2 real data sets (say, microarray data sets). The methods that need to compare include: those discussed in the paper [3,7,13,15,17] if the method is easy to implement as well as support vector machine (SVM) and random Forest.

Fourth, the paper comments that assuming $\Sigma^{-1} (\mu_1 - \mu_0)$ is sparse is weaker than assuming both $\Sigma^{-1}$ and $(\mu_1 - \mu_0)$ are sparse. Mathematically, this is true. However, the more relevant point is that, whether we have an important application example

(say, in genomics or genetics) where $\Sigma^{-1} (\mu_1 - \mu_0)$ is sparse but either $\Sigma^{-1}$ is non-sparse or

$(\mu_1 - \mu_0)$ is non-sparse. If we don't have such application examples in areas many people care, then arguing how

two assumptions are different from each other is largely due to mathematical

curiosity,

not due to scientific interest.

Last, the presentation can be improved. The material can be streamlined so that the main contribution can be more highlighted.
Summary: Good paper in terms of theory, but the authors seems to have very little concerns on practical matters (tuning parameter selection,

computation speed, real-data analysis). This largely down-weights the importance of the paper.

Submitted by Assigned_Reviewer_4

The paper establishes a framework for evaluation the error of linear discriminant approach given conditional Gaussian models. It presents a faster rate of convergence than all existing results without using standard assumptions on separation or low-noise. The paper ends a bit suddenly. A better simulation study and a discussion section would be good additions.

Well written. Clear. Better results.
Summary: It presents a faster rate of convergence than all existing results without using standard assumptions on separation or low-noise. A better simulation study and a discussion section would be good additions.

Author Feedback
Author rebuttal: We thank the reviewers for their kind comments regarding the novelty and importance of our work.
Some reviewers felt that a Conclusion and Discussion section would greatly add to the paper: we will be sure to add this in the camera ready version (including in part the discussions in the responses below).

AR2:

We thank the reviewer for pointing out the related work by Mai and Zou.
It would be interesting to look into the relationship between those various methods and l1 regularized logistic regression in a later work.

AR3:

We thank the reviewer for comments on the presentation of our paper.
We will make relevant edits in the camera ready version.

AR4:

Regarding choosing the tuning parameter (\lambda) in l1 regularized logistic regression: in practice, \lambda is typically chosen using cross-validation or via information-theoretic measures such as BIC. Providing error bounds for such practical selections of lambda is still an active area of research, and certainly an interesting line of future work.

Regarding the tradeoff in computational complexity when using logistic regression in l1 penalty with respect to other related work: We agree that an explicit tradeoff and comparison of computational complexities is an interesting question. It was slightly beyond the focus of this work, which was on statistical rates, but we will definitely expand upon this in an extended version.

Regarding motivating the sparsity assumption from an application perspective: we will do so in the camera-ready version. In particular, many social network and bioinformatics applications have dense connected components in the inverse covariance graph: here, though the inverse covariance graph is not sparse, our less stringent sparsity assumption would still hold.

AR5:

We thank the reviewer for their kind comments on our paper.

AR6:

Regarding the usefulness of our work to machine learning:
(a) In theoretical contributions: we provide a "direct" approach to analyzing 0-1 classification error rates rather than using the traditional approach of bounding the log-likelihood loss, and then using transforms (inequalities), to bound the 0-1 classification error.
(b) In methodological contributions: we systematically studied linear discriminant classifiers under a conditional Gaussian model, and showed that the simple approach of logistic regression with l1-penalty statistically outperforms other linear discriminant approaches, including many *recently proposed* approaches. This has strong consequences for practical users of machine learning based classification techniques.

AR8:

Our paper focuses on the theory of linear discriminant classifiers under conditional Gaussian models, and experiments were only used to demonstrate our theoretical results. In particular, the experiments section presented synthetic simulations whose main-purpose was to show that the bounds we obtain are tight. We plotted the 0-1 error rate against each parameter (sparsity (s), dimension (p), number of samples(n)), and show the stacking of the curves against the control parameter for different (n,p,d) settings.